# Phenotypes and Endotypes of Peach Allergy: What Is New?

**DOI:** 10.3390/nu14050998

**Published:** 2022-02-26

**Authors:** Simona Barni, Davide Caimmi, Fernanda Chiera, Pasquale Comberiati, Carla Mastrorilli, Umberto Pelosi, Francesco Paravati, Gian Luigi Marseglia, Stefania Arasi

**Affiliations:** 1Allergic Unit, Department of Pediatric, Meyer Children’s Hospital, 50139 Florence, Italy; simonabarni@hotmail.com; 2Allergy Unit, CHU de Montpellier, Université de Montpellier, 34295 Montpellier, France; davide.caimmi@gmail.com; 3IDESP, UMR A11–INSERM, Université de Montpellier, 34093 Montpellier, France; 4Department of Pediatrics, San Giovanni di Dio Hospital, 88900 Crotone, Italy; fernandachiera@hotmail.it (F.C.); paravati.f@gmail.com (F.P.); 5Department of Clinical and Experimental Medicine, Section of Pediatrics, University of Pisa, 56126 Pisa, Italy; 6Department of Clinical Immunology and Allergology, I.M. Sechenov First Moscow State Medical University, 119991 Moscow, Russia; 7Department of Pediatrics, University Hospital Consortium Corporation Polyclinic of Bari, Pediatric Hospital Giovanni XXIII, 70124 Bari, Italy; carla.mastrorilli@icloud.com; 8Pediatric Unit, Santa Barbara Hospital, 09016 Iglesias, Italy; umberto.pelosi@gmail.com; 9Department of Pediatrics, University of Pavia, San Matteo Foundation IRCCS Policlinico, 27100 Pavia, Italy; gl.marseglia@smatteo.pv.it; 10Area of Translational Research in Pediatric Specialities, Allergy Unit, Bambino Gesù Children’s Hospital, IRCCS, 00165 Rome, Italy; stefania.arasi@opbg.net

**Keywords:** peach allergy, food allergy, molecular allergy, Pru p 3, Pru p 7, peamaclein, anaphylaxis, oral allergy syndrome, pollen-food allergy syndrome, oral immunotherapy

## Abstract

Peach allergy is emerging as a common type of fresh-fruit allergy in Europe, especially in the Mediterranean area. The clinical manifestations of peach allergy tend to have a peculiar geographical distribution and can range from mild oral symptoms to anaphylaxis, depending on the allergic sensitization profile. The peach allergen Pru p 7, also known as peamaclein, has recently been identified as a marker of peach allergy severity and as being responsible for peculiar clinical features in areas with high exposure to cypress pollen. This review addresses the latest findings on molecular allergens for the diagnosis of peach allergy, the clinical phenotypes and endotypes of peach allergy in adults and children, and management strategies, including immunotherapy, for peach allergy.

## 1. Epidemiology

Peach (*Prunus persica*) is the fruit of *Prunus* trees, belonging to the *Rosaceae* family, including 4828 known species in 104 genera [1,2,3]. In addition to peaches, apples, pears, quinces, apricots, plums, cherries, raspberries, loquats, strawberries, and almonds belong to the *Rosaceae* family [4].

Currently, the peach plant is cultivated in different parts of the world. Peach cultivation is believed to have originated in China and to have been transported, via the silk route, to India, the Middle East, and Persia, before finally spreading towards Europe. China, Italy, Spain, Turkey, and the USA are the leading peach-producing countries [5].

Peach fruit usually ripens between August and September [1]. Peach can be eaten as fresh fruit as well as in treated forms, such as canned, dried, juice, and jam [5].

Peach has been described as a common cause of fresh-fruit allergy in Europe, especially in the Mediterranean area [6]. The prevalence data on fruit allergies are limited, and the available data are derived from scarce studies, especially in children [7,8,9]. In a systematic review conducted by Zuidmeer et al., the overall perceived prevalence of fruit allergies ranged from 0.1% to 4.3% [7]. In particular, 2.2–11.5% of children aged 0–6 years and 0.4–6.6% of adults are affected by fruit allergies, based on self-reported data. One European-based large survey reported the highest and lowest prevalence of allergic sensitization to peach in Germany (11.7%) and Iceland (0.3%), respectively [8]. In another, similar survey, the highest prevalence of peach sensitization was observed in Switzerland (13.4%) and the lowest in Iceland (2.3%) [9]. The overall European prevalence of allergic sensitization to peach increased from 5.4% in 2010 [8] to 7.9% in 2014 [9]. The prevalence data on peach-allergen-specific sensitization have been investigated in Spanish and Italian studies: lipid-transfer protein (LTP) sensitization is predominant in Southern Europe, whereas sensitization to pathogenesis-related 10 (PR-10) is more common in Northern and Central Europe, including areas with Fagales pollen exposure (birch, alder, hazel, hornbeam, oak, beech, and chestnut) [10,11].

Similar to other IgE-mediated food allergies, peach allergy negatively impacts quality of life, causing stress and anxiety. Peach allergy, as with fruit allergies in general, is reported to be associated with less-severe symptoms than food allergies to peanuts and tree nuts; nevertheless, the condition exerts a similar impact on patients’ quality of life: 60% of adults are impacted by fruit allergy in their daily life at home and 62% in their life outside the home [12].

## 2. Peach Allergens

To date, six peach allergens have been recognized [13]. Detailed information on each allergenic protein is provided in Table 1 [14,15,16,17,18,19,20,21,22,23,24,25,26,27,28,29,30,31,32,33,34,35,36,37,38,39].

### 2.1. Pru p 1

Pru p 1 is a member of the PR-10 protein family and is present in the pulp and the skin of peach [14]. It shares a structural homology with the major birch pollen, Bet v1 [15]. For this reason, Pru p1 sensitization is commonly found in Northern and Central Europe, where the exposure to birch pollen is high and usually results in oral allergy syndrome (OAS) symptoms [15]. Pru p1 cross-reacts with other PR-10 protein families, such as *Rosaceae* fruits, hazelnut, carrots, and celery [16]. Pru p 1 is heat-labile and it is found to be sensitive to gastrointestinal digestion [14]. Thus, only the unprocessed form of the fruit leads to the typical symptoms of OAS, whereas cooked peach is tolerated by patients [16].

### 2.2. Pru p 3

Pru p 3 is a non-specific LTP (nsLTP) [17]. The outer surface of the peach (pericarp) has a high concentration of nsLTP [18]. The peel contains seven times higher LTP than the pulp [16]. Pru p 3 cross-reacts with the nsLTP contained in the other fruits of the *Rosaceae* family (apple, plum, cherry and apricot), as well as in vegetables (asparagus, lettuce, tomato, maize, onion, and carrot) and nuts (walnut, hazelnut, almond and peanut) [19]. The LTP nsLTP is a plant panallergen due to its widespread distribution among plant-foods and pollens [16]. The LTPs from different plant-food and pollens can cross-react with each other, causing sensitization and, eventually, symptoms in multiple plant foods, a condition also known as “LTP syndrome” [20]. Pru p 3 is resistant to heat and proteolytic digestion. Therefore, the clinical manifestations of Pru p 3 sensitization can range from mild OAS symptoms to severe systemic allergic reactions (anaphylaxis) [21,22].

### 2.3. Pru p 4

Pru p 4 is a profilin, which is an important pan-allergen, widely found in pollens and vegetables [23]. Pru p 4 is present in the pulp and peel of peach [10]. Pru p 4 cross-reacts with profilins from other members of the *Rosaceae* family (i.e., apple and cherry) and with profilin from unrelated families’ pollen (i.e., *Artemisia vulgaris*, *Betula alba*, *Corylus avellanus*, *P amygdalus*) [24]. Pru p 4 is heat-labile and it can be destroyed by gastrointestinal digestion [25]. For this reason, the usual clinical manifestation of Pru p 4 sensitization is OAS [24,26].

### 2.4. Pru p 7

Pru p 7 is a gibberellin-regulated protein (GRP) [27], also known as Snakin/GASA [28], that was first described by Tuppo et al. in 2013 [27]. Pru p 7 has been found both in the pulp and the peel of the peach [27]. Pru p 7 cross-reacts with several fruits of the *Rosaceae* (i.e., apricot and pomegranate) and *Rutaceae* family (i.e., orange), as well as pollens from the *Cupressaceae* family [29,30,31,32]. Indeed, Pru p 7 sensitivity seems to be most common in areas with high cypress pollen exposure [33]. Pru p 7 is resistant to heat and proteolytic digestion [27]. Thus, the typical allergic symptom of Pru p 7 sensitization is anaphylaxis. Similar to Pru p 3, sensitization to Pru p 7 is considered a risk factor for severe allergic reactions to fresh fruit [33]. Biagioni et al. [34] recently reported the first case series of children with documented Pru p 7 allergies and provided a diagnostic algorithm. The authors suggest performing skin prick tests (SPT) for inhalant and food allergens, including both cypress pollen and Pru p 3-enriched peach peel extracts, in case of a systemic allergic reaction to fruit. In cases of a positive SPT for both cypress- and Pru p 3-enriched peach peel extract and a negative in vitro result for specific IgE (sIgE) to Pru p 3, the diagnosis of Pru p 7 allergy is highly probable. In these cases, whenever possible, determining serum sIgE levels of Pru p 7 is recommended.

### 2.5. Pru p 9

Pru p 9 is a pathogenesis-related protein PR-1, identified in 2018 [13], with a molecular weight of 18 kDa. In 685 Spanish children and adolescents affected by rhino-conjunctivitis and asthma, the sensitization to peach-tree pollen was rated third, after olive tree and grass. Thirty percent (205 out of 685) of children were sensitized to Pru p 9 on skin prick testing [38]. The rate of sensitization to Pru p 9 in children is similar to that in adults from the same area [39]. Pru p 9 is considered a new occupational allergen from peach-tree pollen involved in rhinitis and asthma [39].

## 3. Clinical Manifestations

Similar to other IgE-mediated food allergic reactions, symptoms appear within minutes to two hours from peach ingestion, except for food-dependent exercise-induced anaphylaxis, which can occur up to 4 h later. Reactions can be triggered by the allergen through the oral route, rarely by inhalation or skin contact, and may affect one or more target organs, including the oral mucosa, the skin, the gastrointestinal tract, the respiratory tract, and the cardiovascular system [40,41,42].

Immediate peach-induced reactions could be associated with two clinical patterns: the pollen-food allergy syndrome (PFAS) and a primary food allergy [33].

The clinical manifestations of peach allergy depend on the sensitization profile and, consequently, have a peculiar geographical distribution.

In Northern and Central Europe, peach allergy is mainly secondary to pollen allergy. In this condition, also known as PFAS, pollen allergens are the causative agents of the primary sensitization and food allergy to fruits and vegetables results from cross-reactivity between pollen and food allergens. Conversely, in Mediterranean countries, fruit allergy without related pollinosis is often observed and systemic reactions are frequently reported [43,44].

While, on one hand, it is true that allergy to Pru p 1 is mainly associated with pollen-fruit allergy syndrome, and that Pru p 9 allergy is associated with respiratory symptoms, on the other hand, patients allergic to either Pru p 3 and/or Pru p 7 are at risk of developing severe symptoms, including anaphylaxis and fatal anaphylaxis [33,45,46].

### 3.1. Peach Allergy Secondary to Pollen Allergy

The allergen families involved in peach-induced PFAS include PR10 proteins, profilins, nsLTPs, thaumatin-like proteins, and gibberellin-regulated proteins [47].

PFAS account for up to 60% of food allergies in adult patients and adolescents. It may affect one or more target organs: the skin, the oral mucosa, the gastrointestinal tract, the respiratory tract, and the cardiovascular system [47,48].

The most frequent clinical pattern observed in adult patients and adolescents with PFAS is OAS. Symptoms emerge within 5–15 min of food ingestion and consist of tingling/itching of the lips, tongue, oral mucosa, palate, and throat, with possible mild angioedema associated at the same sites [48].

Most cases resolve spontaneously within 30 min, but 3% of patients present systemic reactions without oropharyngeal symptoms, and 1–8% develop systemic reactions, such as urticaria, dyspnea, wheezing, and anaphylaxis [49,50,51].

Acute generalized urticaria, with or without angioedema, and contact urticaria are the second most frequently observed symptoms of PFAS. Gastrointestinal symptoms, such as nausea, vomiting, abdominal pain, and diarrhea are rarely seen as exclusive manifestations of PFAS. Respiratory symptoms, such as rhinoconjunctivitis, bronchospasm, and laryngeal edema occur more frequently in association with other target organs symptoms rather than in isolation [48].

The presence of comorbidities (atopic dermatitis) and cofactors (exercise, alcohol consumption, use of non-steroidal anti-inflammatory drugs (NSAIDs)) increases the severity of symptoms and the risk of anaphylaxis [50].

### 3.2. Primary Peach Allergy

Primary food allergy to peach, in which the sensitization occurs through the ingestion of the food, is mainly related to nsLTP Pru p 3, although some studies reported primarily airborne sensitization to nsLTPs [52,53].

In the Mediterranean area, there is a high rate of sensitization to nsLTPs, which represents the most frequent cause of both primary food allergy and food-dependent anaphylaxis in adults living in these countries [54,55].

The sensitization to Pru p 3 often occurs early in life. It may be isolated (mono-sensitization) or associated with multiple nsLTP sensitizations, which may lead to multiple plant-food allergies (nsLTP-syndrome) [56].

Pru p 3 sensitization may be asymptomatic or manifest with variable symptom severity, ranging from OAS to anaphylaxis [57,58].

OAS and contact urticaria are the most frequent clinical patterns observed in LTP hypersensitivity. Gastrointestinal symptoms (nausea, vomiting, abdominal pain, diarrhea) may occur as isolated symptoms or in association with the cutaneous, respiratory, or cardiovascular symptoms involved in anaphylaxis [57].

A study on LTP syndrome reported that in a group of 87 patients sensitized to Pru p 3, 44% had anaphylaxis, 43% presented OAS or urticaria, and 13% were asymptomatic. The culprit food belonged to the *Rosaceae* family in 48.8% of the subjects, and the most frequent food involved was peach in both symptomatic groups [59].

Co-sensitization to birch pollen (Bet v 1) and/or to profilin is associated with a lower prevalence of severe reactions and a higher prevalence of local reactions (OAS) [58].

A large prospective study evaluated the phenotype and severity biomarkers of peach-allergic patients sensitized to Pru p 3. The authors showed that most patients were sensitized to other LTP-containing plant foods (LTP syndrome), while only 6.8% were LTP-monoallergic (reacting only to peach and not to other plant foods). Subjects with LTP syndrome had a younger onset of peach allergy, and more asthma and sensitization to Parietaria and profilin than the LTP-monoallergic patients. Anaphylaxis was significantly more frequent in the LTP-monoallergic group, which had no sensitization to profillin. The presence of profilin sensitization was associated with a lower risk of anaphylaxis. No correlation was observed between SPT diameter, Pru p 3 sIgE level, level of nsLTP sensitization, and severity of reaction to peach [60].

Individuals with sensitization to Pru p 3 may develop cross-sensitization to other nsLTPs containing plant foods due to the structural homology between different nsLTPs. Pru p 3 shows a sequence homology from 62% to 81% with analog proteins from apple (Mal d 3), apricot (Pru ar 3), plum (Pru d 3), cherry (Pru av 3), orange (Cit s 3), strawberry (Fra a 3), and grape (Vit v 1). Other LTPs with a structural homology with Pru p 3 are present in peanut (Ara h 9), wheat (Tri a 14), hazelnut (60% with Cor a 8), and walnut (66% with Jug r 3) [61,62]. The risk of cross-reactivity most frequently involves the fruits of the Rosaceae family (apple, plum, apricot, cherry), but also nuts and peanuts. The clinical pattern ranges from local oropharyngeal symptoms up to anaphylaxis [62].

Co-factors are often involved (up to 40% of cases) in clinical expression: fasting, exercise, menstruation, and NSAID could determine the appearance of symptoms in patients sensitized to nsLTPs or influence symptom severity. According to Pascal et al., a cofactor is identified as precipitating anaphylaxis in 32.4% of nsLTPs allergic patients [20].

Sensitization nsLTPs could also be involved in food-dependent exercise-induced anaphylaxis (FDEIA), provoked by the combination of food ingestion and physical exercise within 4 h of food ingestion and within one hour of the start of exercise [63].

In patients with peach-FDEIA, Pru p 3 is the most frequent sensitizer, followed by Pru p 7 [56,63,64].

### 3.3. Peamaclein Allergy

The peach allergen Pru p 7, also known as peamaclein, has recently been identified as a marker of peach allergy severity and as being responsible for peculiar clinical features, sometimes occurring in the presence of cofactors [33,65].

Peamaclein allergy is mostly observed in adolescents and adults. Pru p 7, similarly to Pru p 3, resists heat and digestion and it is suspected to cause a primary food allergy through the gastrointestinal tract route [29]. However, a recent study reported that sensitization to Pru p 7 develops in areas with high exposure to cypress pollen, due to the homology between Cypmaclein and Pru p 7, inducing a PFAS syndrome more severe than those previously described [33,65].

Moreover, Pru p 7 presents homology with Pru m 7 (Japanese apricot), Pun g 7 (pomegranate), Pru av 7 (cherry), and Cit s 7 (orange). In particular, Pru p 7 shows 100% sequence homology with Pru m 7, 97% with Pru av 7, 90% with Pun g 7, 87% with Cit s 7, 84% with black cottonwood GRP, 82% with potato GRP, and 81% with soybean GRP [66]. The clinical cross-reactivity between GRPs was reported among peach, Japanese apricot, orange, and pomegranate. In addition to these fruits, patients with GRP sensitization frequently experience allergic reactions against apple due to the presence of a GRP named applemeclein. It shares a 94% homology with Pru p 7 (peamaclein), Pru m 7 (Japanese apricot), and Pru av 7 (cherry) [67].

A recent multicenter study, including 316 subjects from France, reported that sensitization to Pru p 7 is common in peach-allergic subjects, with a prevalence of 62%, and it occurs often as monosensitization (54%). Furthermore, Pru p 7 sensitization and sIgE levels were higher in patients experiencing Grade 3 reactions, according to EAACI classification [33,68].

Swelling of the face, especially the eyelids, oropharyngeal tightness, and anaphylaxis featured with peamaclein allergy [29].

Inomata et al. observed, among peach-allergic patients sensitized to Pru p 7, that the most frequent symptoms were oropharyngeal (69.2%), followed by laryngeal tightness (46.2%), facial edema (46.2%), eyelid edema (46.2%), urticaria (38.5%), dyspnea (23.1%), nasal obstruction (23.1%), conjunctival injection (15.4%), lip edema (15.4%), loss of consciousness (15.4%), and hypotension (7.7%) [69].

## 4. Diagnosis

As with any diagnostic workup for food allergy, screening allergen-sIgE testing without clinical necessity is discouraged [40,47,70]. A detailed clinical history is therefore crucial for selecting the appropriate confirmatory tests. According to the ICON and EAACI guidelines for food allergies [40], the diagnosis of peach allergy lies on the combination of a convincing clinical history of immediate reaction to peach and positive IgE sensitization testing assessed through SPT to peach (in the form of either extract, molecular components, or fresh peach), and/or IgE sensitization to peach (either extracts or molecular components). Where the diagnosis is unclear, an oral food challenge (OFC) is required as the gold standard test to provide a definitive diagnosis and to prevent patients from unneeded and potentially harmful elimination diets. However, OFC is logistically demanding, and anaphylactic reactions may occur. Reliable prognostic markers or algorithms integrating different clinical and biological parameters for predicting the severity of allergic reactions during OFC are under investigation.

### 4.1. Clinical History

A convincing clinical history is usually defined as one or more immediate reaction(s) within two hours of peach ingestion, inhalation, or direct contact, presenting as acute urticaria or angioedema, contact urticaria, laryngeal swelling, immediate vomiting, rhinitis, cough, wheezing, bronchospasm, hypotension or loss of consciousness, oral allergy syndrome (i.e., itching and tingling of the lips, oral mucosa and/or tongue), or food-dependent exercise-induced anaphylaxis [40,48,70]. The severity of reactions is useful for suspecting specific patterns of sensitization and proper management. Peamaclein (Pru p 7) frequently elicits anaphylaxis [71] and, similarly to allergy to other gibberellins, often includes peculiar clinical symptoms, such as facial swelling and laryngeal tightness, which can be predictive factors for gibberellin allergies [29]. Because of their labile chemical structure, profilins (Pru p 4 in peach) and PR-10 (Pru p 1 in peach) are usually responsible for mild symptoms [72]. Cofactors should be always investigated (e.g., asthma exacerbations, infections, exercise, alcohol, tiredness, use of NSAIDs, and menstruation), since they usually play a crucial role in eliciting reactions in patients allergic to nsLTP (Pru p 3 in peach) and, less frequently, in patients allergic to gibberellins (more evident for Pru m 7 (apricot) and Cit s 7 (orange) less for Pru p 7 (peach)). In t patients who have peach-FDEIA, nsLTPs are the most frequent sensitizers, followed by peamaclein [56].

Therefore, the clinical history should include the following: possible causative food(s) (peach and other fruits/vegetables), the time of onset, the extent and reproducibility of symptoms, the identification of allergic symptoms with plants and plants food(s), the quantity of ingested food, details of the food preparation (e.g., raw vs. cooked, peeled vs. unpeeled), and the relevance of cofactors. 

### 4.2. IgE Sensitization

The use of peach-specific IgE determination combined with clinical history and peach SPT may reduce the need for OFC [73]. Component-resolved diagnosis (CRD), which uses single allergenic components for the assessment of epitope-sIgE, can provide critical information for predicting individualized sensitization patterns and the risk of severe allergic reactions [72]. Only molecular diagnostics makes it possible to identify and differentiate sensitization to LTP or peamaclein. Peach LTP extracts for SPT are contaminated with peamaclein Pru p 7, because LTP Pru p 3 and peamaclein Pru p 7 have similar molecular weights. 

The use of commercial peach extracts for SPT is useful in clinical practice. However, clinicians should consider that peach extract for SPT most likely lacks labile peach allergens (i.e., Pru p 1, and Pru p 4), because these are usually lost during production procedures. By contrast, stable allergens, such as Pru p 3 and Pru p 7, are usually retained in commercial peach extracts [27]. Consequently, SPT with current extracts may furnish a prompt, first-level, component-resolved diagnosis at the bedside [74,75].

The use of serum-sIgE against molecular components provides useful support to the diagnosis and may help with risk stratification, assessment, and management. Pru p 7 is a small protein that is upregulated upon biotic stress. It represents a major allergen associated with severe clinical symptoms and strong cypress pollen sensitization [33]. A study conducted in the southern part of France evaluated 316 patients with suspected peach allergy. According to the ICON and EAACI guidelines for food allergies, peach allergy was diagnosed in 198 subjects. Sensitization to Pru p 7 was present in 171 (54%) of all the subjects in the study and 123 of 198 (62%) were diagnosed as peach-allergic, more than half of whom were sensitized to no other peach allergen. The frequency and magnitude of Pru p 7 sensitization were associated with the presence of a peach allergy, the clinical severity of peach-induced allergic reactions, and the level of cypress pollen exposure. Cypress pollen extract completely outcompeted IgE binding to Pru p 7 [35].

### 4.3. Oral Food Challenge

If the diagnosis of peach allergy is in doubt, OFC is required as it represents the gold standard for the diagnosis of any food allergy. Some OFC protocols are intended to test peach peel and pulp separately [33], others to test them both, and some to assess exercise-induced anaphylaxis [76]. Furthermore, clinicians may consider allergy testing and, ultimately, OFC to plant foods containing nsLTPs or GRP with known potential cross-reactivity with peach if oral tolerance to these foods is in doubt, and according to the patient’s preference (Figure 1).

## 5. Prevention and Management

### 5.1. Primary and Secondary Prevention

To date, no study has shown a possible effective strategy for the primary prevention of peach allergy. Neither polyunsaturated fatty acid supplementation during pregnancy nor the use of probiotics and fish oil supplementation in infancy were effective at preventing the appearance of food allergies [77]. The early introduction of food during diversification could be a possible primary prevention strategy [78]. Even though current data show moderate evidence that the early introduction of peanut and egg reduces the risk of food allergy, there is no sufficient scientific information on other major food allergens [79].

In peach-allergic patients, as with other food allergies, prescribing preventive antihistamines was not shown to be effective at preventing possible allergic reactions; furthermore, this strategy may delay the timely and appropriate use of adrenaline to treat anaphylaxis [40]. The use of mast-cell stabilizers to prevent allergic reactions showed different clinical results, making it not advisable, so far, as a prophylactic strategy for food allergies in general and, therefore, for peach allergy as well [40]. The use of monoclonal antibodies, such as omalizumab and dupilumab, has been suggested instead in the treatment of food allergies, mostly as adjuvant therapy for immunotherapy rather than as a possible preventive strategy against the development of clinical symptoms in allergic patients [80].

### 5.2. Management of Peach Allergy

Once the diagnosis of peach allergy is made, peach should be eliminated from the patient’s diet. Foods possibly cross-reacting with peach allergens should also be investigated by, firstly, assessing whether the patient is exposed to these foods without presenting symptoms and, if this is found not to be the case, by performing skin tests and/or specific IgE dosing. This strategy should mainly be considered for food cross-reacting via Pru p 3 (LTP) or Pru p 7 (peamaclein), given the higher risk of severe reaction associated with sensitization to these allergens.

Management strategies should include both the management of acute accidental reactions and long-term avoidance strategies.

A written emergency action plan for acute reactions should always be provided to all patients with peach allergy. In addition, two adrenaline auto-injectors (AAI) should be prescribed to patients with a history of anaphylaxis to peach.

In order to properly avoid peach, patients should also be educated on how to recognize the presence of peach in commercial products (such as fruit juices). Unfortunately, current labeling practices and legislation do not include the obligation to include the presence of this food, nor to highlight it on the label [81], which could result in the threat of accidental exposure. Other important aspects of educational programs for peach allergy include understanding and recognizing the early signs/symptoms of a possible allergic reaction, avoiding possible triggers or cofactors able to elicit the allergic reaction (e.g., asthma exacerbations, infections, exercise, alcohol, tiredness, use of NSAIDs, and menstruation), and knowing when and how to administer proper treatment, especially if an adrenaline auto-injector has been prescribed [34,40,78].

### 5.3. Allergen Immunotherapy

Immunotherapy is considered an attractive option to treat food allergies and aims at inducing immunological tolerance (the possibility of safe consumption, regardless of regular exposure) of foods.

In terms of oral immunotherapy (OIT), increasing amounts of food are administered to patients with a proven allergy in order to induce desensitization and, possibly, tolerance. In a paper by Patriarca et al. [82], one adult patient underwent OIT with peach and was successfully treated over a 3-month period. Nevertheless, the authors did not provide more specific details on this patient. A more recent study proposed a protocol using peach juice in 24 peach-allergic patients; the protocol followed a sublingual immunotherapy (SLIT) strategy [83]. At the end of the study, the authors were able to administer 200 mL of peach juice to 70.8% of their patients, without reporting severe adverse reactions during the challenge [83]. In any case, peach, as a wholly allergenic source, has not been an allergen on which researchers have focused their attention, as has been the case with OIT. Other SLIT protocols using specific peach proteins have been proposed, such as Pru p 3, on which several studies have been conducted. In the first published study on this topic, after 6 months of SLIT with peach extract quantified in mass units for Pru p 3, 33 patients showed an increase of 3-to-9 fold in their eliciting dose, with a significant difference when compared with the placebo group; moreover, no serious adverse events were reported, and the patients mainly experienced local reactions [84]. A more recent paper confirmed these results on 15 patients, even with an ultra-rush protocol [85]. Furthermore, Beitia et al. showed the effectiveness of Pru p 3 SLIT in a real-life study, including 29 patients, showing that, one year after starting SLIT, 73% had a negative challenge to peach, and, after 2 years, 95% of them did not react to the fruit [86]. In this study, the possibility of using Pru p 3 SLIT to treat patients suffering from LTP syndrome was confirmed, as also shown in other papers, with a positive impact on patients’ quality of life as well [87,88]. Indeed, in the paper by González-Pérez et al., the authors showed that, in 18 adult patients treated for 3 consecutive years with Pru p 3 SLIT, the results on the Food Allergy Quality of Life Questionnaire-Adult Form (FAQLQ-AF) significantly decreased, showing a favorable impact on the patients’ quality of life [87].

Finally, for patients suffering from PFAS, some authors focused on the possibility of treatment with subcutaneous immunotherapy (SCIT), using birch pollen extract. Nevertheless, researchers showed controversial results on this specific issue [89,90,91], and no study was specifically conducted on peach-allergic patients.

In general, even though peach OIT is possibly administered in research and specialized settings, there are currently insufficient data to be able to recommend this approach to treating patients in clinical practice [92].

## 6. Conclusions

Peach allergy may manifest with different clinical symptoms of ranging severity. Based on patients’ sensitization profiles, clinicians may be able to highlight which patients are more at risk of developing a severe allergic reaction. Unfortunately, in clinical practice, clinicians are only able to dose specific serum IgE for the whole peach source, and for Pru p 1 (PR-10), Pru p 3 (LTP), Pru p 4 (profiline), and Pru p 7 (peamaclein). Patients monosensitized to Pru p 9 are known to be at risk of respiratory symptoms, while patients allergic to Pru p 3 and/or Pru p 7 are at risk of experiencing severe allergic reactions. To properly diagnose a peach allergy, therefore, besides presenting a compatible clinical history related to the fruit, patients need to have positive SPT and/or sIgE to available allergens. In cases in which diagnosis cannot be reached by combining these tests, an OFC may be performed, as this procedure is still considered the gold diagnostic standard. Once the diagnosis is made, patients and caregivers should receive proper education on peach avoidance strategies, an emergency action plan for accidental acute reactions and AAIs in case of history of anaphylaxis to peach. OIT is a promising treatment for patients with food allergies who are at high risk of a life-threatening reaction or severe impairment of quality of life. However, currently, peach immunotherapy is not advised in clinical practice.

## Figures and Tables

**Figure 1 nutrients-14-00998-f001:**
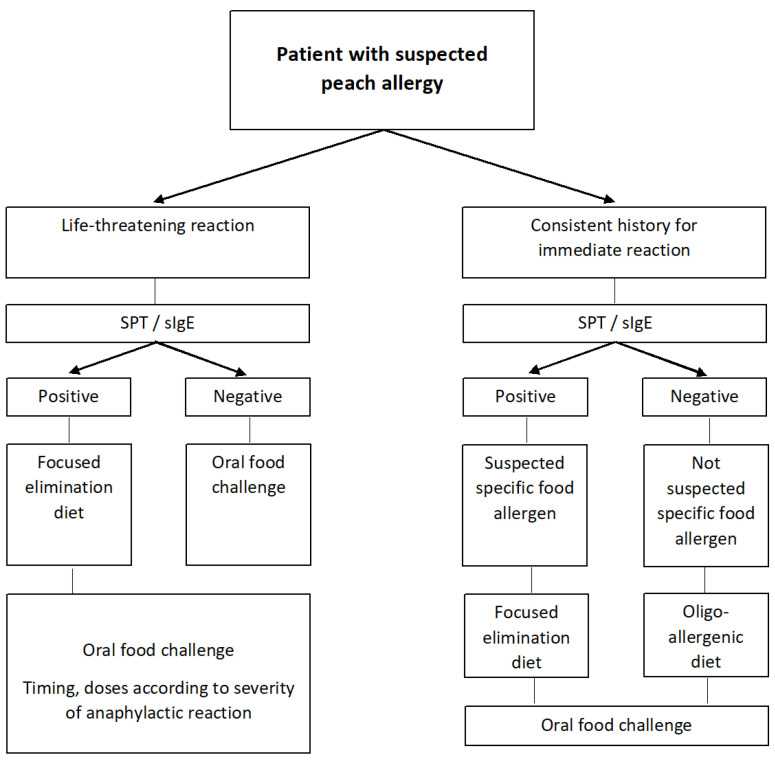
Diagnostic algorithm for peach allergy. Adapted from [40].

**Table 1 nutrients-14-00998-t001:** Main features of peach molecular allergens. Modified from [35].

Allergen	Biochemical Name	Molecular Weight (kDa)	Main Characteristic
Pru p 1	Pathogenesis-related protein group 10, (PR-10), Bet v 1 family member	18	Mainly found in areas with high birch pollen exposure [10].
Pru p 2	Thaumatin-like protein (TLP)	25–28	Pru p 2 from peach was one of the probable allergens causing fruit allergies [36].
Pru p 3	Non-specific lipid-transfer protein 1 (nsLTP1)	10	Major allergen [10]. Present in the outer surface of peach [27]. In total, 54 (96%) out of 57 children showed positive Pru p 3-sIgE in a Spanish study [10].
Pru p 4	Profilin	14	Minor allergen [10]. In total, 52 (12.1%) out of 430 patients were sensitized to profilins in an adult study [37].
Pru p 7	Gibberellin-regulated protein (GRP)	6910.84 Da (Mass spectrometry)	Major allergen [33]. Identified in 2012 [13]. Present both in the pulp and in the peel [27]. Sensitization to Pru p7 was present in 171 (54%) out of 316 subjects with suspected peach allergy [33]. Pru p 7 sensitization was more frequent in peach-allergic (123/198, 62%) than in peach-tolerant (48/118, 41%) patients, *p*-value = 0.0002 [33].
Pru p 9	Pathogenesis-related protein PR-1	18	Identified in 2018 [13]. Sensitization to peach-tree pollen was rated third, after olive tree and grass [38], in areas with peach-tree cultivars. In total, 205 (30%) out of 685 children were sensitized to Pru p 9 on skin prick test [38].

kDa: kilodaltons, IgE: immunoglobulin E; sIgE: specific IgE.

## Data Availability

Not applicable.

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
