# Peer review of "Phenotypes and Endotypes of Peach Allergy: What Is New?"

_nutrients, 2022, doi:10.3390/nu14050998_

Round 1
Reviewer 1 Report
General comments
In the manuscript by Barni S et al entitled “Phenotypes and Endotypes of Peach Allergy: What’s New?”, the authors have made an interesting revision to discuss the latest findings in the diagnosis of peach allergy and management strategies, including specific immunotherapy.
In general, the manuscript is well written, although there are some comments that should be addressed.
Specific comments for revision
Since cross-reactivity between the different peach allergens and other allergens is due to structural similarities, it would be nice to include some indications about the similarity degrees of the allergenic proteins and, therefore, the risk of cross-reactivity between different foods containing them.
During the diagnostic and more specifically in the OFC, in order to make recommendations to the patient, it would be necessary to perform the OFC with other foods that have demonstrated cross-reactivity with peach.
Could the authors propose a diagnostic algorithm?
In the management section, it is indicated the recommendation of eliminating peach from the patient’s diet but it should be also mentioned the indications for other foods that are frequently involved in the clinical symptoms of food allergy due to cross-reactivity.
Pru p 9 has not been sufficiently described along the review, only in the table and conclusion.
In the conclusion, when indicating the capacity of some peach allergens for inducing severe reactions, Pru p 3 should be also mentioned.
Line 224 and 225. Does this +/- mean with or without?
Lines 302-303. The sentence “also, such 302 strategy may delay adrenaline administration of adrenaline”, is a bit confusing, please rephrase.
Author Response
Reviewer 1
Comment 1: In the manuscript by Barni S et al entitled “Phenotypes and Endotypes of Peach Allergy: What’s New?”, the authors have made an interesting revision to discuss the latest findings in the diagnosis of peach allergy and management strategies, including specific immunotherapy. In general, the manuscript is well written, although there are some comments that should be addressed.
Reply 1: Thank you for taking the time to review our manuscript and helping us to improve the quality and the clarity of the paper
Comment 2: Since cross-reactivity between the different peach allergens and other allergens is due to structural similarities, it would be nice to include some indications about the similarity degrees of the allergenic proteins and, therefore, the risk of cross-reactivity between different foods containing them.
Reply 2: We agree. The structural homology and potential cross-reactivity between the different peach allergens and other allergens in plant foods have been further discussed in the text. Four new references (ref. 61,62, 66 and 67) were discussed in the following two new paragraphs (See Lines 210-220 and 239-247 of the revised manuscript).
- «Individuals with sensitization to Pru p 3 may develop cross-sensitization to other nsLTPs containing plant foods, due to structural homology between different nsLTPs. Pru p 3 shows from 62% to 81% of sequence homology with analog protein from apple (Mal d 3), apricot (Pru ar 3), plum (Pru d 3), cherry (Pru av 3), orange (Cit s 3), straw-berry (Fra a 3), grape (Vit v 1). Other LTPs with structural homology to Pru p 3 are present in peanut (Ara h 9), wheat (Tri a 14), hazelnut (60% with Cor a 8) and walnut (66% with Jug r 3 ) [61, 62]. The risk of cross-reactivity most frequently involves the fruits of the Rosaceae family (apple, plum, apricot, cherry) but also nuts and pea-nuts. The clinical pattern ranges from local oropharyngeal symptoms up to anaphy-laxis [62]»
- «Moreover, Pru p 7 presents homology with Pru m 7 (Japanese apricot), Pun g 7 (pomegranate), Pru av 7 (cherry), and Cit s 7 (orange). In particular, Pru p 7 shows 100% of sequence homology with Pru m 7, 97% with Pru av 7, 90% with Pun g 7, 87% with Cit s 7, 84% with black cottonwood GRP, 82% with potato GRP and 81% with soybean GRP [66]. The clinical cross-reactivity between the GRPs was reported among peach, Japa-nese apricot, orange, and pomegranate. In addition to these fruits, patients with GRP sensitization frequently experience allergic reactions against apple due to the presence of a GRP named applemeclein. It shares 94% of homology with Pru p 7 (peamaclein), Pru m 7 (Japanese apricot), and Pru av 7 (cherry) [67]»
Comment 3: During the diagnostic and more specifically in the OFC, in order to make recommendations to the patient, it would be necessary to perform the OFC with other foods that have demonstrated cross-reactivity with peach. Could the authors propose a diagnostic algorithm?
Reply 3: We have implemented the text as follows: «Also, clinicians may consider allergy testing and ultimately OFC to plant foods containing nsLTPs or GRP with known potential cross-reactivity with peach, if oral tolerance to these foods is in doubt and according to the patient’s preference (Figure 1). (See lines 334-340 of the revised manuscript). A Figure with the diagnostic algorithm has been added (see lines 874 or the revised manuscript).
Comment 4: In the management section, it is indicated the recommendation of eliminating peach from the patient’s diet but it should be also mentioned the indications for other foods that are frequently involved in the clinical symptoms of food allergy due to cross-reactivity.
Reply 4: Thank you for such advice. We added the sentence
«Food possibly cross reacting with peach allergens, should also be investigated by firstly assessing if the patient is exposed to this food without presenting symptoms and, if it’s not the case, by performing skin tests and/or dosing specific IgE. Such strategy should mainly be considered for food cross-reacting via Pru p 3 (LTP) or Pru p 7 (peamaclein), given the higher risk of severe reaction associated with sensitization to these allergens» (see Line 369-372 of the revised manuscript)
Comment 5: Pru p 9 has not been sufficiently described along the review, only in the table and conclusion.
Reply 5: We added a new paragraph on Pru p 9: «Pru p 9 is a pathogenesis-related protein PR-1 identified in 2018 [13], with a mo-lecular weight of 18 kDa. In 685 Spanish children and adolescents affected by rhi-no-conjunctivitis and asthma, the sensitization to peachtree pollen was rated third, af-ter olive tree and grass. Thirty percent (205 out of 685) of children were sensitized to Pru p 9 on skin prick test [38]. The rate of sensitization to Pru p 9 in children is similar to that in adults from the same area [39]. Pru p 9 is considered a new occupational al-lergen from peachtree pollen involved in rhinitis and asthma [39]» (see Lines 224-230 of the revised manuscript). We also added and discussed a recent original article on the topic (ref. 39; Somoza, M.L.; et al Subjects develop tolerance to Pru p 3 but respiratory allergy to Pru p 9: A large study group from a peach exposed population. PLoS One 2021)
Comment 6: In the conclusion, when indicating the capacity of some peach allergens for inducing severe reactions, Pru p 3 should be also mentioned.
Reply 6: We agree. As per your suggestion, we amended the paragraph as follows: «Patients monosensitized to Pru p 9 are known to be at risk of respiratory symptoms, while patients allergic to Pru p 3 and/or Pru p 7 are at risk of experiencing severe allergic reaction» (see Line 430-434 of the revised manuscript).
Comment 7: Line 224 and 225. Does this +/- mean with or without?
Reply 7: Thank you. We replaced the symbols “+/-“ with “either or”. The line now reads as: «Combination of a convincing clinical history of immediate reaction to peach and positive IgE sensitization testing assessed through SPT to peach (either extract or molecular components or fresh peach), and/or IgE sensitization to peach (either extracts or molecular components)» (see Line 268-270 of the revised manuscript).
Comment 8: Lines 302-303. The sentence “also, such strategy may delay adrenaline administration of adrenaline”, is a bit confusing, please rephrase.
Reply 8: Agree. We modified the phrase as follows: “ such strategy may delay timely and appropriate use of adrenaline to treat anaphylaxis “ (see Line 356 of the revised manuscript).
Reviewer 2 Report
This review, of Barni et al, provides a thorough review of the current information regarding peach allergy. Still, a refinement of several aspects is required.
Major comments
- Throughout the manuscript, there is a lack of information regarding the harm of peach-allergy to patients' quality of life and function.
Consider inclusion of information from other soueces, such as:
Studies on quality of life of food allergy patients,
Le TM, Lindner TM, Pasmans SG et al. Reported food allergy to peanut, tree nuts and fruit: comparison of clinical manifestations, prescription of medication and impact on daily life. Allergy 2008;63 (7): 910-6 ,
and information from peach-OIT studies that examined quality of life.
Also, Greater emphasis is needed on the risk of peach allergy as a life-threatening allergy.
Lee WJ, Kim DH et al. Targeted temperature management after cardiac arrest with anaphylaxis. Am J Emerg Med 2017; 35(5):807.
- Lines 298-300: Early introduction of foods for prevention of food allergy:
According to current data, there is "moderate certainty of evidence that early introduction of peanut and egg reduce the risk for food allergy development". There is no sufficient information, RCT based, regarding the effect of early introduction of other major food allergens. There is an evidence for preventive benefit of early cow milk introduction on the basis of large interventional studies. This concept is under investigation now, and the statement that: "early introduction…..seems not to be effective", is currently incorrect.
Fleischer DM, Chan ES et al. A consensus approach to the primary prevention of food allergy through nutrition: guidance from the American Academy of Allergy, Asthma, and Immunology; American College of Allergy, Asthma, and Immunology; and the Canadian Society for Allergy and Clinical Immunology. J Allergy Clin Immunol Pract. 2021; 9(1): 22-43
- Lines 375-376 OIT is a promising treatment for patients who are at high risk for a life-threatening reaction or severe impairment of quality of life.
Minor comments
- Line 40: The word "peaches" is redundant.
- Line 338: ….without reporting severe adverse reactions
- Line 343: 33 patients
Author Response
Comment 1: This review, of Barni et al, provides a thorough review of the current information regarding peach allergy. Still, a refinement of several aspects is required.
Reply 1: Thank you for taking the time to review our manuscript and helping us to improve the quality and the clarity of the paper
Comment 2: Throughout the manuscript, there is a lack of information regarding the harm of peach-allergy to patients' quality of life and function. Consider inclusion of information from other soueces, such as: Studies on quality of life of food allergy patients, Le TM, Lindner TM, Pasmans SG et al. Reported food allergy to peanut, tree nuts and fruit: comparison of clinical manifestations, prescription of medication and impact on daily life. Allergy 2008;63 (7): 910-6 , and information from peach-OIT studies that examined quality of life;
Reply 2: Thank you for pointing out this issue. There are not many papers focusing on QoL in patients treated for peach IT, and they focus on SLIT.
We added a paragraph in the introduction (see Lines 64-70 and 414-417 of the revised manuscript) and we developed the issue in the management paragraph as well. We also added 2 new references (n12 and n.87)
Here are the two new paragraphs
- «Peach allergy, as fruit allergy in general, is reported to be associated to less severe symptoms than food allergy to peanuts and tree nuts; nevertheless, such condition show a similar impact on patients’ quality of life: 60% of adults are impacted by fruit allergy in their daily life at home and 62% in their life outside the home [12].»
(see Line 64-70 of the revised manuscript)
- «Indeed, in the paper by González-Pérez et al., the Authors showed that, in 18 adult patients treated for 3 consecutive years with Pru p 3 SLIT, the Food Allergy Quality of Life. Questionnaire-Adult Form (FAQLQ-AF) significantly decreased, showing a favourable impact on their quality of life [87]»
(see Line 414-417 of the revised manuscript)
Comment 3: also, Greater emphasis is needed on the risk of peach allergy as a life-threatening allergy. Lee WJ, Kim DH et al. Targeted temperature management after cardiac arrest with anaphylaxis. Am J Emerg Med 2017; 35(5):807.
Reply 2: We agree with you on the importance of underlying the possible severity of peach allergy. We added therefore a sentence on this topic.
«If, on one hand, it’s true that allergy to Pru p 1 is mainly associated to pollen-fruit allergy syndrome, and to Pru p 9 to respiratory symptoms, on the other hand patients allergic to either Pru p 3 and/or Pru p 7 are at risk of developing severe symptoms, including anaphylaxis and fatal anaphylaxis [33,45,46]» (see Line 153-156 of the revised manuscript). We supported this phrase with two new references on the topic (ref. 45, 46; Lee, W.J.; et al. Targeted temperature management after cardiac arrest with anaphylaxis. Am J Emerg Med 2017, 35(5), 807.; Barradas Lopes, J.; et al. Allergy to lipid transfer proteins (LTP) in a pe-diatric population. Eur Ann Allergy Clin Immunol 2021)
Comment 4: Lines 298-300: Early introduction of foods for prevention of food allergy: According to current data, there is "moderate certainty of evidence that early introduction of peanut and egg reduce the risk for food allergy development". There is no sufficient information, RCT based, regarding the effect of early introduction of other major food allergens. There is an evidence for preventive benefit of early cow milk introduction on the basis of large interventional studies. This concept is under investigation now, and the statement that: "early introduction…..seems not to be effective", is currently incorrect.
Reply 4: Thank you. We modified the sentence, following your indications, to make it more clear on this topic.
«Early introduction of food during diversification, could be a possible primary prevention strategy; even though current data show moderate certainty of evidence that early introduction of peanut and egg reduces the risk of food allergy, there is no sufficient scientific information when considering other major food allergens» (see Line 348-352 of the revised manuscript).
Comment 5: Lines 375-376 OIT is a promising treatment for patients who are at high risk for a life-threatening reaction or severe impairment of quality of life.
Reply 5: As per your request, we added this statement in the conclusion. (see Line 441-443 of the revised manuscript).
Comment 6: Line 40: The word "peaches" is redundant.
Reply 6: Thank you, this was removed.
Comment 7: Line 338: ….without reporting severe adverse reactions
Reply 7: Thank you. As per your suggestion, the word “severe” was added to that phrase.
Comment 8: Line 343: 33 patients
Reply 8: Thank you. As per your suggestion, we specified that “33” were the patients in the active group of the study by Fernández-Rivas M, et al. Allergy. 2009 Jun;64(6):876-83.
Round 2
Reviewer 2 Report
The manuscript is now improved by the additions/revisions